# A causal relationship between sarcopenia and cognitive impairment: A Mendelian randomization study

Hengzhi Liu[1,2]☯, Yi Fan[3,4]☯, Jie Liang[5,6], Aixin Hu[5,6], Wutong Chen[7], Hua Wang[5,6], Yifeng Fan[7], Mingwu Li[1,2], Jun Duan[1,2], Qinzhi Wang[1,2]*

1 Department of Orthopaedics, Huangshi Central Hospital, Huangshi, China, 2 Department of Orthopaedics, Affiliated Hospital of Hubei Polytechnic University, Huangshi, China, 3 Department of Infection, Huangshi Central Hospital, Huangshi, China, 4 Department of Infection, Affiliated Hospital of Hubei Polytechnic University, Huangshi, China, 5 Department of Orthopaedics, The First College of Clinical Medical Science, China Three Gorges University, Yichang, China, 6 Department of Orthopaedics, Yichang Central People's Hospital, Yichang, China, 7 Department of Orthopaedics, China Three Gorges University, College of Basic Medical Sciences, Yichang, China

☯ These authors contributed equally to this work.
* wangqinzhi2024@163.com

**Data Availability Statement:** All relevant data are within the manuscript and its Supporting Information files.

## Abstract

### Objective

Sarcopenia and cognitive impairment often coexist in the elderly. In this study, we investigated the causal relationship between sarcopenia-related muscle characteristics and cognitive performance.

### Methods

We used linkage disequilibrium score regression (LDSC) and Mendelian Randomization (MR) analyses to estimate genetic correlations and causal relationships between genetically predicted sarcopenia-related muscle traits and cognitive function, as well as cognitive function-based discovery samples and replicated samples. Estimated effect sizes were derived from a fixed-effects meta-analysis.

### Results

Our univariate genome-wide association study (GWAS) meta-analysis indicated a causal relationship between appendicular lean mass (ALM) ($\beta$ = 0.049; 95% confidence interval (CI): 0.032–0.066, P < 0.001) and walking pace ($\beta$ = 0.349; 95% CI: 0.210–0.487, $P$ < 0.001) with cognitive function, where a causal relationship existed between ALM in both male and female ($\beta_{ALM-Male(M)}$ = 0.060; 95% CI: 0.031–0.089, $P_{ALM-M}$ < 0.001; $\beta_{ALM-Female(F)}$ = 0.045; 95% CI: 0.020–0.069, $P_{ALM-F}$ < 0.001) with cognitive function. Low grip strength was not causally associated with cognitive function ($\beta$ = -0.045; 95% CI: -0.092 - -0.002, $P$ = 0.062). A reverse causality GWAS meta-analysis showed a causal relationship between cognitive function and ALM ($\beta$ = 0.033; 95% CI: 0.018–0.048, $P$ < 0.001) and walking pace ($\beta$ = 0.039; 95% CI: 0.033–0.051, $P$ < 0.001), where ALM in both male and female showed a

**Funding:** The author(s) received no specific funding for this work.

**Competing interests:** The authors have declared that no competing interests exist.

causality ($\beta_{ALM-M}$ = 0.041; 95% CI: 0.019–0.063, $P_{ALM-M}$ < 0.001; $\beta_{ALM-F}$ = 0.034; 95% CI: 0.010–0.058, $P_{ALM-F}$ = 0.005). Cognitive function was not causally related to low grip strength ($\beta$ = -0.024; 95% CI: -0.073–0.025, $P$ = 0.344). Multivariable MR1 (MVMR1) analyses showed a significant causal relationship for ALM ($\beta$ = 0.077; 95% CI: 0.044–0.109, $P$ = 0.000) and walking pace ($\beta$ = 0.579; 95% CI: 0.383–0.775, $P$ = 0.000) and cognitive function. Multivariable MR2 (MVMR2) multivariate analysis showed that ALM causality remained ($\beta$ = 0.069; 95% CI: 0.033–0.106, $P$ = 0.000), and walking pace ($\beta$ = 0.589; 95% CI: 0.372–0.806, $P$ = 0.000).

## Conclusions

Bidirectional two-sample MR demonstrated that sarcopenia-related muscle characteristics and cognitive performance were positive causal genetic risk factors for each other, while a multivariable MR study demonstrated that low ALM and a slow walking pace were causally involved in reduced cognitive performance. This study suggests a causal relationship between sarcopenia and cognitive impairment in older adults and provide new ideas for prevention and treatment.

## 1. Introduction

Sarcopenia is a syndrome represented by age-related skeletal muscle mass loss, accompanied by decreased muscle strength and reduced physical performance [1]. Muscle-related decline in strength and function causes many adverse outcomes, such as increased risk of falls, fractures, frailty, disability, and high mortality [2]. Currently, sarcopenia prevalence in individuals over 60 years old is 10%–12% worldwide. Globally, approximately 50 million individuals suffer from sarcopenia, and it is expected to increase to 200 million by 2050, which will generate more global health problems and increase economic burdens [3,4]. Muscle mass is usually assessed by ALM; dual-energy X-ray absorptiometry scanning is a low cost, low radiation technology and is used to measure total muscle mass in arms and legs [5,6]. Grip strength and walking pace are simple and effective muscle strength and screening assessments for sarcopenia [7–9].

Cognitive impairment is a chronic disease associated with aging, and is characterized by a decline in memory, thinking, perception, language, decision-making, planning, and reasoning, ultimately leading to dementia [10–12]. At the same time, sarcopenia and cognitive impairment have become important chronic diseases that jeopardize health in elderly individuals, leading to global problems such as reduced quality of life, death, and an increased health care burden; thus, disease prevention and management are global issues that require urgent attention.

Several observational clinical studies have reported that sarcopenia is associated with an increased risk of cognitive impairment, regardless of the geographical location of the study population and the sarcopenia definition [13,14].

Patients with sarcopenia in low- and middle-income countries have faster cognitive decline rates and an increased risk of cognitive impairment [15]. sarcopenia-related muscular traits represented by a slow walking pace and low grip strength can be used to predict short- and long-term cognitive decline in older adults [16]. A longitudinal study of 6,435 middle-aged and older Korean adults, followed for 6 years, found that low grip strength increased the risk

of developing cognitive impairment by 36% [17]. Another national survey of 2,540 individuals aged 60 years and older reported a significant association between low ALM and cognitive impairment [18]. However, these studies did not fully include sarcopenia-related muscular characteristics, and even one study reported no associations between these characteristics and cognitive impairment [19]. Therefore, due to conflicting evidence and the inherent limitations of observational studies, causal relationships between sarcopenia and cognitive impairment remain unclear.

MR The use of genetic variants that are strongly associated with exposure to determine the causality of exposure and outcome reflects the effective effect of exposure on long-term effective effects on outcome. Genetic variants are randomly assigned at conception; they do not change with disease onset and course, they minimize the influence of external confounders and various biases on outcomes, and they are somewhat immune to small sample limitations [20,21]. Therefore, we investigated genetic correlations and causality between genetically predicted sarcopenia-related muscle traits and cognitive impairment using LDSC and MR. By clarifying the causal relationships between sarcopenia and cognitive impairment, we can guide and develop public health policy to reduce the risk of individuals developing both disorders.

## 2. Methods

### 2.1. Data sources

In our analyses, we used pooled data from published and publicly available GWAS. We used muscle characteristics associated with sarcopenia, including ALM, low hand grip strength (measured for individuals aged 60 years or older), and walking pace. Genetic associations for ALM were derived from a GWAS of 450,243 UK Biobank (UKB) participants. Of these, 205,513 and 244,730 were of male and female European ancestry, respectively [22]. Jamar J00105 hydraulic hand dynamometer measured grip strength was assessed according to the EWGSOP definition of low grip strength < 30 kg in men and grip strength < 20 kg in females. Low hand grip strength was extracted from a GWAS meta-analysis that included 256,523 Europeans from 22 independent cohorts [23]. Walking pace was recorded as a category phenotype using the ACE touchscreen: 'slow', 'steady/average' or 'brisk', with slow walking pace defined as < 3 miles/ hour, steady/average walking pace as 3–4 miles/hour, and brisk walking as > 4 miles/hour. Genetic associations for walking pace were obtained from a GWAS of 459,915 UKB participants. General cognitive performance scores were used to assess cognitive performance; the higher the score, the greater the cognitive performance, and vice versa. Cognitive performance data were obtained from the SSGAC consortium, which included 257,841 Europeans, and was the largest published public GWAS dataset on cognitive performance [24]. Exposures and outcomes were characterized without sample overlap, and all study participants were of European descent. The cognitive functions used to replicate the cohort came from the 'within family GWAS consortium', including 22,593 Europeans (**S1 Table in S1 File**).

### 2.2. Single nucleotide polymorphisms (SNPs) in exposure and outcome selection

The genome-wide significance parameter for exposed instrumental variable (IV) SNPs was P < 5e−8. Because too many SNPs were included (526 SNPs), the genome-wide significance parameter for IV SNPs, with ALM as the exposed SNP, was P < 5e−20. Previous MR studies used more stringent thresholds in identifying too many SNPs [25]. For IV-associated SNPs, a linkage disequilibrium test, set at kb > 10 MB (R2 < 0.001), was performed to ensure the mutual independence of selected genetic variants and exclude palindromic SNPs with

intermediate allele frequencies. IVs that were significantly correlated with an outcome ($P < 5e$ $-5$) were excluded. Steiger tests were used to indicate no reverse causality for IVs [26]. F-value $= (n-k-1/k) (R^2/1-R^2)$ [27], where N = the GWAS sample size and K = the number of variant instruments. In principle, an F-value $> 10$ was chosen for analysis, indicating a less likelihood of weak IV bias [28]. $R^2 = 2 \times \beta^2 \times (1-EAF) \times EAF$ [29], where EAF represented the effect allele frequency and $\beta$ represented the estimated genetic effect of each SNP. $R^2$ reflected the extent to which IVs explained the exposure.

LDSC was used to estimate genetic correlations between traits based on GWAS summary statistics. Genetic correlations between sarcopenia-related traits and cognitive function were assessed using cross-trait LDSC [30–32].

MVMR was also performed; in MVMR1 we used ALM, low grip strength, and walking pace to build a model to determine if one or more characteristic was responsible for cognitive impairment risk. In MVMR1 models, each with sarcopenia-related traits no interference by any other factors or adjustment. MR before study, smoking is associated with cognitive function and sarcopenia-related muscle characteristics have a causal relationship [33,34]. Therefore, in MVMR2, smoking was considered a confounding factor. We selected a population data source with the largest sample size for smoking (smoking was derived from the GSCAN consortium, and MVMR2 was used to determine whether characteristics associated with sarcopenia remained causally related to cognitive function when smoking effects were adjusted). In reverse MR analysis, cognitive function was treated as exposure and its causal effect on muscle characteristics associated with sarcopenia was assessed.

Bilateral P values $< 0.05$ were considered significant. We further adjusted the threshold for the number of exposure phenotypes using the Bonferroni correction approach. Thus, the threshold for statistical significance for the three sarcopenia-related muscle characteristic exposures in inverse variance weighting (IVW) was set at $P < 0.05/3 = 0.017$. A value of $0.017 < P < 0.05$ was suggestive of a potential genetic association. MR analyses were performed using MR-PRESSO (v.1.0), two-sample MR packages [35], Radial MR (v.1.0), and R software (v.9.2.19).

## 2.3. Statistical analysis

IVW was the main univariable analysis method as it provided the most accurate causal estimation [36]. MR-Egger [37], weighted median (WM) [38], and weighted mode methods [39] were used as complementary methods and sensitivity analyses. Although precision and efficiency were relatively low, MR-Egger regression generated estimates by correcting for pleiotropy, and the intercept was used to check for pleiotropy. WM provided robust and consistent effect estimates when 50% of inheritance was for valid IVs [37,40].

Heterogeneity was tested using Cochran Q heterogeneity tests [41,42]. Sensitivity analysis was performed using leave-one-out analysis to explore the effects of individual SNPs on causal associations. Radial IVW was used to remove IVs that contributed significantly to heterogeneity [43]. Scatter plots showed pleiotropy results calculated using MR-Egger intercept tests. Sensitivity analysis was performed using leave-one-out analysis to explore the effect of individual SNPs on causal associations.

## 3. Ethics approval

Published GWAS studies had passed ethical review at the time of study.

## 4. Results

For all genetic associations used in bidirectional two-sample MR analyses of sarcopenia and cognitive performance, each SNP is listed (**S2 Table in S1 File**). IVs with high heterogeneity

**Table 1. Genetic correlations between sarcopenia and cognitive impairment.**

| trait | cognitive performance | | | cognitive function | | |
|---|---|---|---|---|---|---|
| | $r_g$ | SE | P-Value | $r_g$ | SE | P-Value |
| ALM | 0.153 | 0.017 | **0.000** | 0.216 | 0.037 | **0.000** |
| ALM-M | 0.129 | 0.019 | **0.000** | 0.187 | 0.040 | **0.000** |
| ALM-F | 0.168 | 0.019 | **0.000** | 0.239 | 0.040 | **0.000** |
| low hand grip strength | -0.066 | 0.030 | 0.030 | -0.122 | 0.061 | 0.046 |
| walking pace | 0.315 | 0.023 | **0.000** | 0.410 | 0.047 | **0.000** |

ALM: Appendicular lean mass; ALM-M: Appendicular lean mass-Male; ALM-F: Appendicular lean mass-Female; SE: Standard error.

were removed by Radial MR (**Supplementary Figures K1–20 in S1 Data**). All F-statistic ranges for IVs exceeded 10, indicating no weak instrumental bias. MR-Steiger tests were used to test each SNP to verify the correctness of the causal direction (**S1 Table in S1 File**).

### 4.1. LDSC regression analyses

These analyses were performed to assess genetic associations between sarcopenia-related muscle characteristics and cognitive function. Analyses showed a causal relationship between ALM and walking pace and cognitive function, and a suggestive association between low grip strength and cognitive function (**Table 1**).

### 4.2. The effects of sarcopenia on cognitive performance (forward)

Univariate discovery analyses showed that ALM had a causal relationship ($\beta = 0.046$; 95% confidence interval (CI): 0.028–0.064, $P = 0.000$), and walking pace ($\beta = 0.357$; 95% CI: 0.210–0.503, $P = 0.000$) with cognitive function, both male and female of ALM is a causal relationship ($\beta_{ALM-M} = 0.050$; 95% CI: 0.018–0.082, $P_{ALM-M} = 0.002$; $\beta_{ALM-F} = 0.043$; 95% CI: 0.017–0.070, $P_{ALM-F} = 0.001$) and cognitive function. According to Bonferroni correction analyses, no causal relationship was identified between low grip strength and cognitive function ($\beta = -0.061$; 95% CI: 0.112–0.010, $P = 0.019$). Replication cohort results showed that ALM had a causal relationship ($\beta = 0.078$; 95% CI: 0.020–0.135, $P = 0.008$) and cognitive function, low grip strength ($\beta = 0.055$; 95% CI: -0.073 to -0.183, $P = 0.401$), and walking pace ($\beta = 0.278$; 95% CI: of -0.147–0.704, $P = 0.200$) had no causal relationship with cognitive function. Found that the results of analysis and replication queue is not very consistent, thus obtained the final estimate GWAS meta-analysis. A causal connection between ALM ($\beta = 0.049$; 95% CI: 0.032–0.066, $P < 0.001$), and walking pace ($\beta = 0.349$; 95% CI: 0.210–0.487, $P < 0.001$) with cognitive function. The causal relationship was observed in both male and female ALM ($\beta_{ALM-M} = 0.060$; 95% CI: 0.031–0.089, $P_{ALM-M} < 0.001$; $\beta_{ALM-F} = 0.045$; 95% CI: 0.020–0.069, $P_{ALM-F} < 0.001$) and cognitive function. No causal relationship was identified between low grip strength and cognitive function ($\beta = -0.045$; 95% CI: -0.092 - -0.002, $P = 0.062$) (**Fig 1**). Furthermore, we did not check for heterogeneity and pleiotropy in ALM and walking pace with MR Egger intercept and Cochran's Q test (**S3 Table in S1 File**).

### 4.3. The effects of cognitive performance on sarcopenia (reverse)

Reverse causal discovery analysis showed that cognitive function was associated with ALM ($\beta = 0.054$; 95% CI: 0.029–0.079, P = 0.000) and walking pace ($\beta = 0.073$; 95% CI: 0.059–0.088, $P = 0.000$). Among them, cognitive function was causally associated with ALM in both male and female ($\beta_{ALM-M}$: 0.062; 95% CI of 0.019 to 0.105, $P_{ALM-M} = 0.005$; $\beta_{ALM-F}$: 0.084; 95% CI of

| Exposure(outcome) | No.of SNP | Method | Beta(95% CI) | P |
|---|---|---|---|---|
| Univariable(forward) | | NA | | |
| ALM(cognitive performance) | 155 | IVW | 0.046 (0.028 to 0.064) | 0.000 |
| Low hand grip strength(cognitive performance) | 11 | IVW | −0.061 (−0.112 to −0.010) | 0.019 |
| Walking pace(cognitive performance) | 28 | IVW | 0.357 (0.210 to 0.503) | 0.000 |
| Univariable(reverse) | | NA | | |
| Cognitive performance(ALM) | 76 | IVW | 0.054 (0.029 to 0.079) | 0.000 |
| Cognitive performance(Low hand grip strength) | 115 | IVW | −0.073 (−0.145 to −0.001) | 0.048 |
| Cognitive performance(Walking pace) | 94 | IVW | 0.073 (0.059 to 0.088) | 0.000 |
| Multivariable(524SNPs) | | IVW | | |
| ALM(cognitive performance) | 524 | IVW | 0.077 (0.044 to 0.109) | 0.000 |
| Low hand grip strength(cognitive performance) | 524 | IVW | 0.022 (−0.026 to 0.070) | 0.371 |
| Walking pace(cognitive performance) | 524 | IVW | 0.579 (0.383 to 0.775) | 0.000 |

(axis: 0    0.2    0.4    0.6    0.8)

**Fig 1. Sarcopenia causality and cognitive impairment in univariable and meta-analysis models.**

0.081 to 0.141, $P_{ALM-F}$ = 0.000). No causal relationship was identified between cognitive function and low grip strength ($\beta$ = -0.073; 95% CI: - 0.145 - -0.001, $P$ = 0.048). In the replication cohort, cognitive function was significantly associated with ALM ($\beta$ = 0.022; 95% CI: 0.004–0.041, $P$ = 0.018) and low grip strength ($\beta$ = 0.019; 95% CI: -0.048–0.086, $P$ = 0.585), while walking pace ($\beta$ = 0.018; 95% CI: 0.010–0.033, $P$ = 0.001). Among them, the causal relationship between male and cognitive function in ALM, female ALM no causal relationship ($\beta_{ALM-M}$ = 0.034; 95% CI: 0.009–0.060, $P_{ALM-M}$ = 0.008; $\beta_{ALM-F}$ = 0.011; 95% CI: -0.018–0.039, $P_{ALM-F}$ = 0.468). Discovery analysis and replication cohort results were not quite consistent, which led to a final estimate from GWAS meta-analysis that cognitive function was causal for ALM ($\beta$ = 0.033; 95% CI: 0.018–0.048, $P$ <0.001), and walking pace ($\beta$ = 0.039; 95% CI: 0.033–0.051, $P$ < 0.001), among them, both male and female of ALM is a causal relationship ($\beta_{ALM-M}$ = 0.041; 95% CI: 0.019–0.063, $P_{ALM-M}$ < 0.001; $\beta_{ALM-F}$ = 0.034; 95% CI: 0.010–0.058, $P_{ALM-F}$ = 0.005). No causal relationship was identified between cognitive function and low grip strength ($\beta$ = -0.024; 95% CI: 0.073–0.025, P = 0.344) (**Fig 1**). Meanwhile, we did not detect heterogeneity using the MR Egger intercept and Cochran's Q test. (**S4 Table in S1 File**).

Bidirectional two-sample MR visualization results are presented as follows. IVW funnel plots were roughly symmetrical, with no significant outliers identified (**Supplementary Figures S1–20 in S1 Data**). Forest plots reflected results estimated after association by relevant individual SNPs using the Wald ratio method (**Supplementary Figures T1–20 in S1 Data**). Scatter plots reflected estimated effect sizes for sarcopenia and cognitive performance phenotypes (**Supplementary Figures R1–20 in S1 Data**). IVW MR analyses of SNPs were individually eliminated using the leave-one-out method (**Supplementary Figures H1–20 in S1 Data**).

## 4.4. The effects of sarcopenia on cognitive performance (multivariable)

As the three muscle traits associated with sarcopenia were interrelated, we performed MVMR1 to determine whether they were causally related to cognitive function, independent of other traits. We observed that IVW ultimately included 524 SNPs, and multivariate analyses showed significant causal associations between ALM ($\beta$ = 0.077; 95% CI: 0.044–0.109, $P$ = 0.000), walking pace ($\beta$ = 0.579; 95% CI: 0.383–0.775, $P$ = 0.000), and cognitive function. When smoking was included in MVMR2, IVW ultimately included 480 SNPs, and multivariate analyses showed that ALM causality remained ($\beta$ = 0.069; 95% CI: 0.033–0.106, $P$ = 0.000) and also walking pace ($\beta$ = 0.589; 95% CI: 0.372–0.806, $P$ = 0.000) (**Fig 2**).

| Exposure(outcome) | No.of SNP | Method | Beta(95% CI) | P |
|---|---|---|---|---|
| MVMR1(524SNPs) | | NA | | |
| ALM(cognitive performance) | 524 | IVW | 0.077 (0.044 to 0.109) | 0.000 |
| Low hand grip strength(cognitive performance) | 524 | IVW | 0.022 (−0.026 to 0.070) | 0.371 |
| Walking pace(cognitive performance) | 524 | IVW | 0.579 (0.383 to 0.775) | 0.000 |
| MVMR2(480SNPs) | | NA | | |
| ALM(cognitive performance) | 480 | IVW | 0.069 (0.033 to 0.106) | 0.000 |
| Low hand grip strength(cognitive performance) | 480 | IVW | 0.023 (−0.030 to 0.076) | 0.398 |
| Walking pace(cognitive performance) | 480 | IVW | 0.589 (0.372 to 0.806) | 0.000 |
| Smoking | 480 | IVW | −0.076 (−0.142 to −0.010) | 0.024 |

0     0.2     0.4     0.6     0.8

**Fig 2. Sarcopenia causality and cognitive impairment in multivariable models.**

## 5. Discussion

In this study, we investigated genetically predicted causal relationships between sarcopenia and cognitive performance using large-scale GWAS summary data. In univariable MR, ALM and walking pace GWAS summary data representing sarcopenia were significantly causally related to cognitive performance, with lower ALM and slower walking pace likely associated with lower cognitive performance scores and a higher risk of cognitive impairment. Because Bonferroni correction analyses were used to avoid false positives, no significant causal relationship was identified between low hand grip strength GWAS summary data and cognitive performance, as confirmed by multivariable analyses. In reverse MR analyses, a significant causal relationship was identified between cognitive performance, ALM, and walking pace, with lower cognitive performance scores possibly representing lower ALM and a slower walking pace in sarcopenia. These associations between muscle characteristics and cognitive impairment provide new insights; for the first time, muscle mass and walking pace, rather than strength, are the main factors associated with cognitive performance.

Several observational studies have reported associations between sarcopenia and cognitive impairment. In their prospective study, Salinas-Rodríguez et al. [44] demonstrated that annual mild cognitive impairment rates in older non-sarcopenia adults was 0.8%, whereas in older adults with sarcopenia it was 1.5%. In a follow-up study of 2,982 elderly individuals, Hu et al. showed that mild cognitive impairment incidences in a non-sarcopenia group, a possible sarcopenia group, and a sarcopenia group were 10.1%, 16.5%, and 24.2%, respectively [45].

The specific mechanisms underlying associations between sarcopenia and cognitive impairment have not been elucidated; however, some common pathways exist between the two. Firstly, elevated inflammatory levels such as interleukin-6, tumor necrosis factor-α, and C-reactive protein affect protein homeostasis, thereby inducing a catabolic state and muscle atrophy [46,47]. Secondly, mitochondrial dysfunction, oxidative stress, chronic inflammation, and hormones such as glucocorticoids, sex steroids, thyroid hormones, growth hormones, and insulin-like growth factor-1, are involved in protein anabolism and apoptotic molecular damage. These molecules diminish as an individual ages and may eventually lead to cognitive impairment and sarcopenia [48,49]. Both conditions also share multiple risk factors, e.g., age, obesity, cardiovascular disease, diabetes, and decreased activity [50].

Sarcopenia is an independent risk factor for cognitive impairment [51,52]. Reduced muscle mass in sarcopenia may inhibit myocytokine production, which mediates the muscle–brain axis and helps improve cognitive performance in the brain; therefore, myocytokine deficiency may trigger cognitive impairment [53]. Also, skeletal muscle tissue contraction releases brain-

derived neurotrophic factor which regulates synapses in the brain, and its deficiency during sarcopenia is associated with neurodegenerative processes [54].

Cognitive impairment is an independent risk factor for sarcopenia [55,56]. Firstly, behavioral changes, reduced activity, and dysphagia secondary to cognitive impairment accelerate muscle loss and decrease muscle strength, which may lead to sarcopenia [57,58]. Secondly, oxidative stress associated with cognitive impairment disrupts protein synthesis and catabolism, leading to mitochondrial dysfunction and apoptosis, which in turn predispose individuals to developing sarcopenia [59]. Additionally, abnormal neurotransmitter levels and activity in the central nervous system, and also insufficient oxygen supply to the brain in patients with cognitive impairment, can reduce muscle activity and lead to sarcopenia [60].

Our study had several limitations. Firstly, results from other MR methods (MR- Egger, WM, and weighted mode) were not fully consistent with the IVW method. However, since MR analysis is primarily an IVW method, IVW results were preferred in the absence of pleiotropy. Secondly, SNP selection in GWAS data may have increased sample overlap rates between exposure and outcome, thus biasing outcomes; however, selecting GWAS data from as many different samples as possible, as well as F-values much larger than 10, can minimize sample overlap [61]. Thirdly, due to GWAS database limitations, gender, height, weight, ethnicity, and underlying disease data can be restricted, and also, as our main study was conducted in a European population, the data may not be generalizable to other populations.

## Supporting information

**S1 File.**
(XLSX)

**S1 Data.**
(ZIP)

## Author Contributions

**Conceptualization:** Yi Fan, Jie Liang, Hua Wang, Qinzhi Wang.

**Data curation:** Hengzhi Liu, Yifeng Fan, Mingwu Li.

**Formal analysis:** Hengzhi Liu.

**Project administration:** Qinzhi Wang.

**Resources:** Jun Duan, Qinzhi Wang.

**Software:** Hengzhi Liu, Yi Fan, Wutong Chen.

**Supervision:** Jie Liang, Aixin Hu.

**Visualization:** Hengzhi Liu, Yi Fan.

**Writing – original draft:** Hengzhi Liu, Yi Fan, Aixin Hu.

**Writing – review & editing:** Hengzhi Liu, Hua Wang.

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
