## [Decision Letter · Decision Letter 0]

29 Apr 2024

PONE-D-24-06716The causal relationship between sarcopenia and cognitive impairment: a Mendelian randomization studyPLOS ONE

Dear Dr. Liang,

Thank you for submitting your manuscript to PLOS ONE. After careful consideration, we feel that it has merit but does not fully meet PLOS ONE’s publication criteria as it currently stands. Therefore, we invite you to submit a revised version of the manuscript that addresses the points raised during the review process.

We look forward to receiving your revised manuscript.

Kind regards,

Sayani Das, PhD

Academic Editor

PLOS ONE

Journal Requirements:

Reviewers' comments:

Reviewer's Responses to Questions

**Comments to the Author**

1. Is the manuscript technically sound, and do the data support the conclusions?

Reviewer #1: Partly

2. Has the statistical analysis been performed appropriately and rigorously? 

Reviewer #1: No

3. Have the authors made all data underlying the findings in their manuscript fully available?

Reviewer #1: Yes

4. Is the manuscript presented in an intelligible fashion and written in standard English?

Reviewer #1: Yes

5. Review Comments to the Author

Reviewer #1: Thank you very much for giving me the opportunity to review this manuscript. I am very interested in this topic and have read it carefully. I have some questions for the author to answer.

First, the authors mention that they used multivariate Mendelian randomization, but I don't seem to find a specific description of what potential confounders were controlled for by MVMR in the paper.

Second, I recommend that the authors use linkage disequilibrium score regression to explore genetic correlations between phenotypes. This is the theoretical basis of the study.

Furthermore, there are significant sex differences in sarcopenia related phenotypes, which may also be associated with cognitive function. Therefore, gender stratification analysis is necessary. As far as I know, gender-stratified GWAS full summary data is currently available.

Can the authors validate their results with data from other sources?

Finally, I suggest that the author revise the article through a native speaker and correct any errors. I also suggested that the author rewrite the introduction, which I felt was lacking in the context. Among them, the introduction of why this study was conducted was not convincing.

As a reminder, it appears that the submitter of the manuscript mistakenly entered the corresponding author's affiliate in plos one's system.

6. PLOS authors have the option to publish the peer review history of their article (what does this mean?). If published, this will include your full peer review and any attached files.

Reviewer #1: No

---

## [Author Response · Author response to Decision Letter 0]

14 Jun 2024

Reviewer 1#

First, the authors mention that they used multivariate Mendelian randomization, but I don't seem to find a specific description of what potential confounders were controlled for by MVMR in the paper.

The author’s answer:

As the three muscle traits associated with sarcopenia were interrelated, we performed MVMR1 to determine whether they were causally related to cognitive function, independent of other traits. We observed that IVW ultimately included 524 SNPs, and multivariate analyses showed significant causal associations between ALM (β = 0.077; 95% CI: 0.044–0.109, P = 0.000), walking pace (β = 0.579; 95% CI: 0.383–0.775, P = 0.000), and cognitive function. When smoking was included in MVMR2, IVW ultimately included 480 SNPs, and multivariate analyses showed that ALM causality remained (β = 0.069; 95% CI: 0.033–0.106, P = 0.000) and also walking pace (β = 0.589; 95% CI: 0.372–0.806, P = 0.000) (Figure 2).

Figure 2. Sarcopenia causality and cognitive impairment in multivariable models

Reviewer 2#

Second, I recommend that the authors use linkage disequilibrium score regression to explore genetic correlations between phenotypes. This is the theoretical basis of the study.

The author’s answer:

LDSC was used to estimate genetic correlations between traits based on GWAS summary statistics. Genetic correlations between sarcopenia-related traits and cognitive function were assessed using cross-trait LDSC (1-3)

These analyses were performed to assess genetic associations between sarcopenia-related muscle characteristics and cognitive function. Analyses showed a causal relationship between ALM and walking pace and cognitive function, and a suggestive association between low grip strength and cognitive function (Table 1).

Table 1.The genetic correlations between sarcopenia and cognitive impairment

Reference: 1. Bulik-Sullivan BK, Loh P-R, Finucane HK, Ripke S, Yang J, Patterson N, et al. LD Score regression distinguishes confounding from polygenicity in genome-wide association studies. Nat Genet. 2015;47(3):291-5.

2. Bulik-Sullivan B, Finucane HK, Anttila V, Gusev A, Day FR, Loh P-R, et al. An atlas of genetic correlations across human diseases and traits. Nat Genet. 2015;47(11):1236-41.

3. Gazal S, Finucane HK, Furlotte NA, Loh P-R, Palamara PF, Liu X, et al. Linkage disequilibrium-dependent architecture of human complex traits shows action of negative selection. Nat Genet. 2017;49(10):1421-7.

Reviewer 3#

Furthermore, there are significant sex differences in sarcopenia related phenotypes, which may also be associated with cognitive function. Therefore, gender stratification analysis is necessary. As far as I know, gender-stratified GWAS full summary data is currently available.

The author’s answer:

Genetic associations for ALM were derived from a GWAS of 450,243 UK Biobank (UKB) participants. Of these, 205,513 and 244,730 were of male and female European ancestry, respectively (4).This analysis has been supplemented here. No gender-stratified GWAS data were found for the remaining low grip strength and walking pace.

Reference:4. Pei Y-F, Liu Y-Z, Yang X-L, Zhang H, Feng G-J, Wei X-T, et al. The genetic architecture of appendicular lean mass characterized by association analysis in the UK Biobank study. Commun Biol. 2020;3(1):608.

Reviewer 4#

Can the authors validate their results with data from other sources?

The author’s answer:

Cognitive functions used for the replication cohort were obtained from the Within family GWAS consortium and included 22,593 Europeans. Furthermore, GWAS meta analyses, thus obtained the final estimate of GWAS meta analysis. Bidirectional two-sample MR demonstrated that sarcopenia-related muscle characteristics and cognitive performance were positive causal genetic risk factors for each other, while a multivariable MR study demonstrated that low ALM and a slow walking pace were causally involved in reduced cognitive performance.

Figure 1. Sarcopenia causality and cognitive impairment in univariable and meta-analysis models

Reviewer 5#

Finally, I suggest that the author revise the article through a native speaker and correct any errors. I also suggested that the author rewrite the introduction, which I felt was lacking in the context. Among them, the introduction of why this study was conducted was not convincing.

The author’s answer:

Below are credentials for revising the article and correcting any errors by native speakers.Founded in 2003 in the USA, BioMed Proofreading® is a leading English language copyediting company that focuses on copyediting, paraphrasing/rephrasing and translation of academic manuscripts in life sciences. 

MR The use of genetic variants that are strongly associated with exposure to determine the causality of exposure and outcome reflects the effective effect of exposure on long-term effective effects on outcome. Genetic variants are randomly assigned at conception; they do not change with disease onset and course, they minimize the influence of external confounders and various biases on outcomes, and they are somewhat immune to small sample limitations (5, 6). Therefore, we investigated genetic correlations and causality between genetically predicted sarcopenia-related muscle traits and cognitive impairment using LDSC and MR. By clarifying the causal relationships between sarcopenia and cognitive impairment, we can guide and develop public health policy to reduce the risk of individuals developing both disorders.

Reference:5. Carter AR, Sanderson E, Hammerton G, Richmond RC, Davey Smith G, Heron J, et al. Mendelian randomisation for mediation analysis: current methods and challenges for implementation. Eur J Epidemiol. 2021;36(5):465-78.

6. Gao Q, Hu K, Yan C, Zhao B, Mei F, Chen F, et al. Associated Factors of Sarcopenia in Community-Dwelling Older Adults: A Systematic Review and Meta-Analysis. Nutrients. 2021;13(12).

---

## [Decision Letter · Decision Letter 1]

7 Aug 2024

A causal relationship between sarcopenia and cognitive impairment: a Mendelian Randomization study

PONE-D-24-06716R1

Dear Dr. WANG,

We’re pleased to inform you that your manuscript has been judged scientifically suitable for publication and will be formally accepted for publication once it meets all outstanding technical requirements.

Kind regards,

Francesco Curcio, M.D., Ph.D.

Academic Editor

PLOS ONE

Additional Editor Comments (optional):

No further comments

Reviewers' comments:

Reviewer's Responses to Questions

**Comments to the Author**

1. If the authors have adequately addressed your comments raised in a previous round of review and you feel that this manuscript is now acceptable for publication, you may indicate that here to bypass the “Comments to the Author” section, enter your conflict of interest statement in the “Confidential to Editor” section, and submit your "Accept" recommendation.

Reviewer #1: All comments have been addressed

Reviewer #2: All comments have been addressed

2. Is the manuscript technically sound, and do the data support the conclusions?

Reviewer #1: Yes

Reviewer #2: Yes

3. Has the statistical analysis been performed appropriately and rigorously? 

Reviewer #1: Yes

Reviewer #2: Yes

4. Have the authors made all data underlying the findings in their manuscript fully available?

Reviewer #1: Yes

Reviewer #2: Yes

5. Is the manuscript presented in an intelligible fashion and written in standard English?

Reviewer #1: Yes

Reviewer #2: Yes

6. Review Comments to the Author

Reviewer #1: The authors have made a detailed revision in response to my comments, and I believe that the revised version is acceptable for publication in plos one. And I don't have any more questions.

Reviewer #2: The manuscript is improved. All questions are answered. No further comments are nedeed. The manuscript merits to be published in PONE.

7. PLOS authors have the option to publish the peer review history of their article (what does this mean?). If published, this will include your full peer review and any attached files.

Reviewer #1: No

Reviewer #2: No

---

## [Editor Report · Acceptance letter]

28 Aug 2024

PONE-D-24-06716R1 

PLOS ONE

Dear Dr. Wang, 

I'm pleased to inform you that your manuscript has been deemed suitable for publication in PLOS ONE. Congratulations! Your manuscript is now being handed over to our production team.

Kind regards, 

on behalf of

Dr. Francesco Curcio 

Academic Editor

PLOS ONE